# Low-Precision Arithmetic for Fast Gaussian Processes

**Wesley J. Maddox**[1,*]  **Andres Potapczynski**[1,*]  **Andrew Gordon Wilson**[1]

[1]Center for Data Science, New York University
[*]Equal contribution.

## Abstract

Low-precision arithmetic has had a transformative effect on the training of neural networks, reducing computation, memory and energy requirements. However, despite its promise, low-precision arithmetic has received little attention for Gaussian process (GP) training, largely because GPs require sophisticated linear algebra routines that are unstable in low-precision. We study the different failure modes that can occur when training GPs in half precision. To circumvent these failure modes, we propose a multi-faceted approach involving conjugate gradients with re-orthogonalization, mixed precision, and preconditioning. Our approach significantly improves the numerical stability and practical performance of conjugate gradients in low-precision over a wide range of settings, enabling GPs to train on 1.8 million data points in 10 hours on a single GPU, without requiring any sparse approximations.

## 1 INTRODUCTION

Low-precision computations can profoundly improve training time, memory requirements, and energy consumption. Moreover, as hardware accelerators continue to develop, the scalability gains of low-precision algorithms will continue to appreciate into the future. However, training in low-precision can increase the magnitude of round-off errors and decrease the supported range of the numerical representation, leading to lower quality gradients and ultimately sacrificing performance or, in the worst cases, convergence of the training procedure.

Recent works have developed efficient and stable methods for training deep neural networks in low-precision to great effect, often via the development of new floating point representations [Intel, 2018, Wang and Kanwar, 2019], mixed precision representations [Micikevicius et al., 2017, Das et al., 2018, Yang et al., 2019], or alternative summation techniques [NVIDIA, 2017, Micikevicius et al., 2017, Zamirai et al., 2021].

Gaussian processes (GPs) are significantly more expensive than neural networks, and thus potentially stand more to gain from low-precision computations. GPs typically require $\mathcal{O}(N^3)$ computations and $\mathcal{O}(N^2)$ memory, for $N$ data points. However, GPs typically require sophisticated algebraic operations such as Cholesky decompositions for solving linear systems, which suffer badly from round-off error and are very poorly suited to half precision. Developing these approaches for low-precision computation is an open area of research [Higham and Pranesh, 2021].

Instead, we focus on iterative techniques for Gaussian processes [Gibbs and MacKay, 1997, Cutajar et al., 2016, Gardner et al., 2018] that use conjugate gradients to compute the solutions to linear systems while exploiting extremely efficient MapReduce matrix vector multiplications (MVMs) using KeOps [Charlier et al., 2021, Feydy et al., 2020]. These approaches are particularly appealing in the context of low-precision computations: (1) they suffer less from round-off errors than Cholesky decompositions [Gardner et al., 2018]; (2) they are easy to parallelize on GPUs, which are being designed specifically to accelerate low-precision computations [Gardner et al., 2018, Wang et al., 2019]; (3) they reduce memory consumption, which is a major *computational* bottleneck, since lower memory consumption reduces the number of computationally expensive matrix partitions [Wang et al., 2019].

While there are many challenges to overcome, we demonstrate that training GPs in half precision can be stable, efficient, and practical. In particular:

- We numerically show that half precision kernel matrices have qualitatively different eigenspectra than kernels in higher precisions, and represent covariances at much shorter distances.

*Accepted for the 38th Conference on Uncertainty in Artificial Intelligence* (UAI 2022).

- We empirically investigate the effect of kernel choice, CG tolerance, preconditioner rank, and compact support on stability in half precision.

- Based on these insights, we propose a new numerically stable conjugate gradients (CG) algorithm that converges quickly in half precision, particularly due to the use of re-orthogonalization, mixed precision and the logsumexp trick.

- We provide a powerful practical implementation of Gaussian processes in half precision that reduces training times on 1.8 million data point datasets to less than 10 hours on a single GPU, a major advance over recent CG milestones of 1 million data points in 3 days on 8 GPUs [Wang et al., 2019].

- We release code at `https://github.com/AndPotap/halfpres_gps`.

## 2 PRELIMINARIES

We briefly introduce background on different floating point precision representations, and on Gaussian processes. We also review how solving linear systems can be viewed as an optimization problem, and present conjugate gradients as an accelerated linear solver.

### 2.1 NUMERICAL PRECISION

Modern computer architectures represent numbers in floating point precision, often with either 32-bit (single) or 64-bit (double) precision [Kahan, 1996]. *Single precision* uses 1 bit for the sign of the number ($s$), 8 bits for the exponent ($e$), and 23 bits for the mantissa (or significant, $m$). Then, the value of a number is represented as

$$\text{value}\,(s, e, m) = (-1)^s \times 2^{e-127} \times 1.m$$

where $s \in \{0, 1\}$, $e \in \{0, \cdots, 255\}$ and $m\{1, \cdots, 1 + \sum_{i=1}^{23} 2^{-i}\}$. *Single precision* can then represent values from $\pm \left(2 - 2^{-23}\right) \times 2^{127} \approx \pm 3.4028235 \times 10^{38}$ and measure small numbers, in absolute value, of the order of $2^{-126} \approx 1.175 \times 10^{-38}$. The round-off error is approximately of $2^{-24} \approx 6 \times 10^{-8}$ (Chapter 2.5, Watkins [2010]).

In contrast, *half precision*, a representation with 16-bits, uses 1 bit for the sign, 5 bits for the exponent and 10 bits for the mantissa (or significant). Using again $s, e$ and $m$ to represent the previous bits, we obtain the following representation

$$\text{value}\,(s, e, m) = (-1)^s \times 2^{e-14} \times 1.m$$

where $s \in \{0, 1\}$, $e \in \{0, \cdots, 255\}$ and $m\{1, \cdots, 1 + \sum_{i=1}^{10} 2^{-i}\}$. Immediately we see that the range of values decreases to $\pm \left(2 - 2^{-10}\right) \times 2^{17} \approx \pm 2.620 \times 10^5$ and that the lowest value that can be represented is $2^{-14} \approx$

$6.1035 \times 10^{-5}$. The round-off error is now significantly higher as well, on the order of $2^{-14} \approx 10^{-4}$. Much modern hardware attempts to reduce the round-off error via fused multiply-and-accumulate (FMAC) compute units to accumulate operations at higher precisions [NVIDIA, 2017, Zamirai et al., 2021, Micikevicius et al., 2017]. We also note the existence of the bfloat16 standard [Intel, 2018], which uses a slightly different representation, preserving the range of single precision, while using fewer bits in the mantissa, thereby trading off the range for increased round-off error.

### 2.2 GAUSSIAN PROCESSES

We consider regression tasks on observed data $\mathcal{D} = \{(\mathbf{x}_i, y_i)\}_{i=1}^N$ for $\mathbf{x}_i \in \mathbb{R}^D$ and $y_i \in \mathbb{R}$. The Gaussian process (GP) model for the data is

$$f\,(\cdot) \sim \mathcal{GP}\,(m\,(\cdot), k\,(\cdot, \cdot)),$$
$$y_i = f\,(\mathbf{x}_i) + \epsilon_i, \quad \epsilon_i \sim \mathcal{N}\,(0, \sigma^2)$$

where $k_\theta\,(\cdot, \cdot)$ is the covariance kernel, $m\,(\cdot)$ is the mean function (in the next equations we assume it set to zero without loss of generality) [Rasmussen and Williams, 2008]. We train the kernel hyperparameters, $\theta$, by minimizing the negative log marginal likelihood:

$$\begin{aligned}\mathcal{L}\,(\boldsymbol{\theta}) &= -\log p\,(\mathbf{y}|\mathbf{X}; \boldsymbol{\theta}) \\ &= \frac{1}{2}\log\left|\tilde{\mathbf{K}}\right| + \frac{1}{2}\mathbf{y}^T\tilde{\mathbf{K}}^{-1}\mathbf{y} + \frac{N}{2}\log(2\pi)\end{aligned} \quad (1)$$

where $\tilde{\mathbf{K}} \in \mathbb{R}^{N \times N}$ is the Gram matrix of all data points with diagonal observational noise

$$\tilde{\mathbf{K}}_{i,j} = (\mathbf{K} + \sigma^2 I)_{i,j} = k\,(\mathbf{x}_i, \mathbf{x}_j) + \sigma^2 \mathbb{I}_{i=j},$$

representing the evaluated covariance matrix as $\mathbf{K}$. To perform hyperparameter estimation of $\boldsymbol{\theta}$, we need to compute the gradient of the loss function, which is given by

$$\begin{aligned}\nabla_{\boldsymbol{\theta}}\mathcal{L} &= \frac{1}{2}\text{Tr}\left(\tilde{\mathbf{K}}^{-1}\nabla_{\boldsymbol{\theta}}\tilde{\mathbf{K}}\right) - \frac{1}{2}\mathbf{y}^T\tilde{\mathbf{K}}^{-1}\left(\nabla_{\boldsymbol{\theta}}\tilde{\mathbf{K}}\right)\tilde{\mathbf{K}}^{-1}\mathbf{y} \\ &\approx \frac{1}{M}\sum_{i=1}^M \mathbf{z}_i^\top\tilde{\mathbf{K}}^{-1}\nabla_{\boldsymbol{\theta}}\tilde{\mathbf{K}}\mathbf{z}_i - \frac{1}{2}\mathbf{y}^T\tilde{\mathbf{K}}^{-1}\left(\nabla_{\boldsymbol{\theta}}\tilde{\mathbf{K}}\right)\tilde{\mathbf{K}}^{-1}\mathbf{y}\end{aligned}$$
$$(2)$$

where the first line is the result of classical matrix derivatives [Rasmussen and Williams, 2008] and the second line comes from the application of Hutchinson's trace estimator [Gibbs and MacKay, 1997, Gardner et al., 2018]. The limiting time complexity is that of computing $\mathbf{v} = \tilde{\mathbf{K}}^{-1}\mathbf{y}$, which costs $\mathcal{O}(N^3)$ operations and space when using the Cholesky decomposition [Rasmussen and Williams, 2008]. Instead, Gibbs and MacKay [1997] and then Gardner et al. [2018], proposed to use conjugate gradients (CG) to compute these linear systems, reducing the time complexity down to $\mathcal{O}(JN^2)$, for $J$ steps of conjugate gradients.

## 2.3 CONJUGATE GRADIENTS

We use conjugate gradients (CG) to solve the linear system $\mathbf{A}\mathbf{x} = \mathbf{b}$ for $\mathbf{x}$, so that $\mathbf{x} = \mathbf{A}^{-1}\mathbf{b}$ [Hestenes and Stiefel, 1952, Nocedal and Wright, 2006, Demmel, 1997]. CG uses a three-term recurrence where each new term requires only a matrix-vector multiplication with $\mathbf{A}$. Formally, each CG iteration computes a new term of the following summation: $\mathbf{A}^{-1}\mathbf{b} = \sum_{i=1}^{N} \alpha_i \mathbf{d}_i$, where, for simplicity, we assume that the algorithm is initialized at $\mathbf{x}_0 = \mathbf{0}$, $\alpha_i$ are the step-size coefficients and $\mathbf{d}_i$ are the orthogonal conjugate search directions [Golub and Loan, 2018]. In infinite precision representation, $N$ iterations of CG produce all the $N$ summation terms and recover the exact solution. The CG algorithm is shown in Algorithm 1 in the Appendix.

## 3 RELATED WORK

**Iterative Gaussian Processes**   There has been extensive research on alternative algorithms to reduce the cubic runtime complexity of training Gaussian processes via Cholesky. In this work, we primarily focus on iterative GPs methods, prioritizing over other alternative approaches. Gibbs and MacKay [1997] studied iterative techniques for GPs by using conjugate gradients to solve the linear systems that result from training GPs and also to estimate the stochastic trace term that appears when computing the loss. In contrast to Cholesky, these iterative approaches reduce the overall complexity of GP regression to $\mathcal{O}(JN^2)$ for $J$ steps of CG. Cutajar et al. [2016] re-visited these approaches, and applied preconditioners based on low-rank kernel approximations to increase the convergence speed to the linear system solutions, using double precision. Gardner et al. [2018] proposed the batch conjugate gradients algorithm that we extend in this paper and used Lanczos to estimate log determinants, focusing their efforts in single precision. Wang et al. [2019] extended this work and enabled exact GPs to be trained on 1 million data points by using 8 GPUs over 3 days via data partitioning. Meanti et al. [2020] also used KeOps over several GPUs for Nyström-based kernel regression, achieving results on datasets of up to 1 billion data points.

**Lower Precision Arithmetic**   Interest in training neural network models in lower precisions than the traditional double or single precisions has been around as early as Hammerstrom [1990]. In the modern era, Gupta et al. [2015] and Chen et al. [2014] pioneered the usage of lower precision arithmetic to reduce memory costs so that larger deep neural network models could be trained. Gupta et al. [2015] proposed the usage of stochastic rounding to reduce errors from the loss of precision when moving to half (e.g. 16-bit arithmetic) precision down from single precision, while Chen et al. [2014] used special representations to enable accurate training of DNNs. These works have led to a flurry

of research into *mixed precision* training of deep neural networks [Micikevicius et al., 2017, Das et al., 2018, Yang et al., 2019]. In mixed precision training, the activations and gradients of each DNN layer are propagated in half but the weights are stored in single precision [Micikevicius et al., 2017]. The success of these lower precision approaches has led to significant efforts in reducing the memory overhead even further via quantization of neural network layers [Jacob et al., 2018, Das et al., 2018], the development of specialized chip architectures that speed up lower precision arithmetic such as modern GPUs [NVIDIA, 2017] and TPU cores [Wang and Kanwar, 2019], and new standards that enhance mixed precision training such as bfloat16 [Wang and Kanwar, 2019, Intel, 2018]. The use of lower precision arithmetic has found some applications outside of deep learning for kernel approximations. For example, using quantized random Fourier features as in [Zhang et al., 2019, Li and Li, 2021] or inside of structured basis functions like Fastfood [Le et al., 2013, Yang et al., 2015].

**Improving CG convergence**   When using infinite precision arithmetic, CG is guaranteed to converge relatively fast to the solution of the linear system. However, the round-off error introduced when using finite precision affects the convergence to the solution. There are several strategies that can generally improve the stability and convergence of CG, such as: preconditioning, re-orthogonalization, mixed precision and blocked arithmetic. For double precision GP inference, Cutajar et al. [2016] found that preconditioning was necessary for the CG solves to be accurate, a finding echoed by Gardner et al. [2018] and more recently Wenger et al. [2021]. Gardner et al. [2018], Wang et al. [2019] and Wenger et al. [2021] proposed using pivoted Cholesky preconditioners [Harbrecht et al., 2012]. Additionally, recent work in numerical methods, such as Gratton et al. [2021], has argued for the use of re-orthogonalization in CG as a general numerical strategy. Haidar et al. [2017] and Haidar et al. [2018] alternatively proposed iterative refinement inside of GMRES (a method related directly to CG) for solving dense linear systems on GPUs in lower precision arithmetic, while Abdelfattah et al. [2020] considered mixed precision solves using iterative refinement. Higham [2021] and Higham and Pranesh [2021] argue for blocked arithmetic to reduce round-off errors, which we demonstrate has considerable advantages in Figure 4b.

## 4 WHAT HAPPENS TO KERNEL MATRICES IN HALF PRECISION?

In this section, we investigate the properties of half precision kernels, finding that they have a truncated support, that their eigenspectra are qualitatively different than in single precision (leading to different generalization properties) and that direct methods such as the standard Cholesky decomposition fails in half precision due to round-off error. In

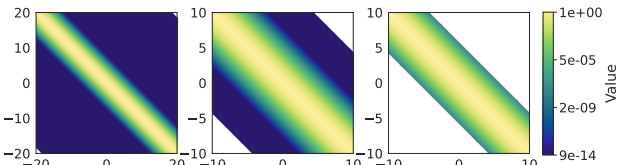

Figure 1: Elementwise logarithms of RBF kernel matrices in double (left), single (middle), and half (right) precisions. As we move to lower precision, the support of the kernel matrices becomes increasingly truncated, which can be exploited for faster inference. White values show where the kernel matrix is exactly zero, producing a kernel with compact support. A compactly supported kernel produces more efficient kernel matrix MVMs as we can safely remove data points that have distances from a given test point larger than can be represented in a given numerical precision.

Section 5 we leverage these insights to develop effective methods for low-precision GP inference.

## 4.1 FINITE PRECISION KERNELS HAVE FINITE SUPPORT

First, in Figure 1, we display the elementwise logarithms of RBF kernel matrices of size 500 as we lower the precision from double (left) to single (middle) to half (right) precision. The elementwise logarithm allows us to see that there are significantly more values of the kernel matrix represented as *exactly zero*, displayed as white, for the half precision kernels than for the higher precisions. For the half and single precision kernels, we consider data evenly spaced in $[-10, 10]$ while for double we consider data in $[-20, 20]$.

We can exactly quantify the range of support of common stationary kernel matrices for a given lengthscale. For example, for a Matérn-$1/2$ kernel with lengthscale, $l$, the kernel function will be exactly zero when $k(x, x') = k(d) = \exp\{-d/l\} < \epsilon$ (satisfied when $d > -\log \epsilon/l$), where $\epsilon$ is the smallest representable value in that given precision. We provide analogous results for other kernels in the Appendix. If we have $d = 0.75$ (equivalent to $x = -0.5$ and $x' = 0.25$), then we cannot represent a relationship in half precision between the two points for lengthscales greater than about $9.24 \approx -\log \epsilon/0.75$. We show the maximum distance representable in double, single and half precisions for RBF (Figure 2a) and rational quadratic kernels (Figure 2b, displaying 5 units (a proxy for the maximum distance of input data standardized to have mean 0 and variance 1) as a gray dashed line. RBF kernels have the smallest support, especially for short lengthscales, followed by Matérn kernels (see Figure A.1b), and then rational quadratic kernels.

We can further understand these results by examining the spectrum of the eigenvalues of RBF kernels. In infinite precision, an RBF kernel has support over the full space, with

$\lambda_k > 0$ for each $k$. For $k \geq \mathcal{O}(\log \delta)$ with $\delta$ representing the round-off error in the specified precision, then $\lambda_k = 0$ in finite precision, empirically reducing the support of the kernel (see Appendix D for further details). These results are reminiscent of how probabilities of a Gaussian variable taking values several standard deviations from its mean are numerically zero, due to the sharp decay of the tails of the distribution.

Kernels represented with compact support can be exploited for scalable computations. Indeed, we demonstrate the potential for this type of structure exploitation when computing GP predictive means in Figure 2c; first, we compute the predictive mean cache $\mathbf{v} = \tilde{\mathbf{K}}^{-1}\mathbf{y}$ before either computing a full predictive mean $\mu_{\text{full}} = K(x^*, \mathbf{X})\mathbf{v}$ or a truncated predictive mean by dropping all data points that will be represented as zero in half precision, e.g. $\mu_{\text{trunc.}} = K(x^*, \mathbf{X}_{\text{close}})\mathbf{v}_{\text{close}}$, which drops approximately $3/4$ of the data in this example. As shown in Figure A.1c, the error is zero, indicating the predictive mean does not change at all!

## 4.2 INVESTIGATING THE EIGENSPECTRUM

We next investigate the eigenvalue spectrum across precisions, considering a Matérn-$1/2$ kernel, and data drawn from $[-3, 3]$, displaying the empirical spectrum in Figure 2d. First, we see that the condition number of the linear system is about the same across precisions, since the maximum eigenvalue is the same (see Figure A.2), and we know that for GP kernel matrices the smallest eigenvalue is really close to the noise hyperparameter value $\sigma^2$. Second, we see that the eigenvalues of the half precision kernel decay slower before dropping to zero when compared to other precisions. As the per iteration progress in CG is bounded by the difference between the current iteration eigenvalue and the first (Thm 5.5 of Nocedal and Wright [2006]), then we would expect CG to take more iterations to converge in half precision as the progress of each step in CG is minimal if the eigenvalues are similar [Demmel, 1997].

Furthermore, the difference between the eigenvalue spectrum of single and half precision may lead to worse generalization. This point can be argued through the effective dimension of a kernel matrix $\mathbf{K}$ defined as $N_{\text{eff}}(\mathbf{K}, \sigma^2) := \sum_{i=1}^{N} \frac{\lambda_i}{\lambda_i + \sigma^2}$ where $\lambda_i$ represents the eigenvalues of $\mathbf{K}$. First, note that the effective dimension is a critical term in generalization bounds of kernel regression and GP methods [e.g., Zhang, 2005, Opper and Vivarelli, 1998]. These generalization bounds are bounded above by $N_{\text{eff}}(\mathbf{K}, \lambda)/N$; therefore, higher effective dimensionality leads to looser bounds. We prove in Appendix D that

$$\mathbb{E}\left(\sum_{i=1}^{N} \frac{Q(\lambda_i)}{Q(\lambda_i) + s}\right) \geq N_{\text{eff}}(\mathbf{K}, s), \qquad (3)$$

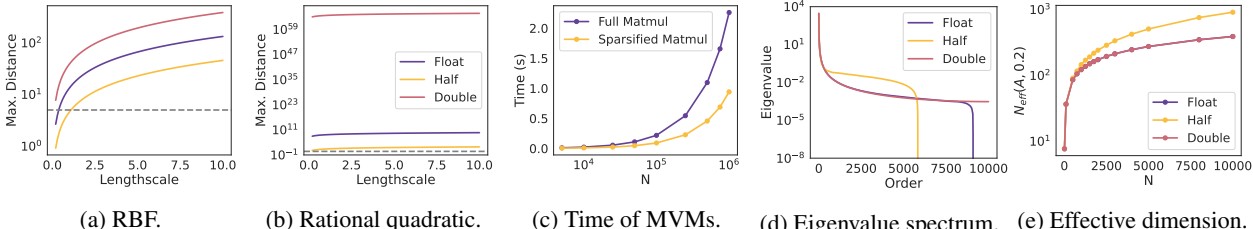

| (a) RBF. | (b) Rational quadratic. | (c) Time of MVMs. | (d) Eigenvalue spectrum. | (e) Effective dimension. |

Figure 2: **(a, b)** Maximum representable distance across precisions for RBF and rational quadratic (RQ, $\alpha = 5$) kernels. Shown as a gray line for reference is 5 units of distance. RBF kernels have significantly shorter representable distances than RQ kernels. **(c)** Time for both the full and truncated MVMs as the dataset size grows. The truncated MVM grows at a significantly lower rate than the full MVM because it only operates on one quarter of the full matrix. **(d)** An observed eigenvalue spectrum for a Matérn-$1/2$ kernel across precisions; the kernel evaluated in half precision has a much slower rate of decay in its spectrum than either single or double, which produces a much larger effective dimensionality **(e)** across matrix sizes.

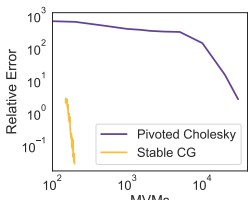

Figure 3: Pivoted Cholesky fails to solve a linear system in half, whereas preconditioned CG solves the system efficiently and accurately.

where $Q(\lambda_i)$ denotes the quantization error of $\lambda_i$ and, following Li and De Sa [2019], we assume that $Q(\lambda_i)$ is distributed uniformly $U(\lambda_i - \delta, \lambda_i + \delta)$, where $\delta$ represents the round-off error of our numerical precision. We then argue in Appendix D how the LHS of equation 3 increases for lower precisions (that is, for higher $\delta$) leading to provably worse generalization. Second, it has been empirically shown how for similar training losses, lower effective dimensionality correlates with better generalization [Zhang, 2005, Caponnetto and Vito, 2007, Maddox et al., 2020]. In Figure 2e we empirically observe that the slower eigenvalue decay of half precision implies that, as $N$ increases, the effective dimensionality increases much faster for the half precision kernel. This finding is consistent with our previous theoretical analysis and also suggests that half precision will have a worse generalization than single precision in the context of kernel methods.

### 4.3 WHAT ABOUT DIRECT METHODS?

Can direct methods, such as Cholesky decompositions, effectively solve linear systems in low-precision? Support for Cholesky factorizations in half precision is not included in GPU based linear algebra libraries (LAPACK) and approaches for half precision Cholesky factorizations use iterative techniques on the backend [Higham and Pranesh, 2021].

We studied the effectiveness of direct methods by using Eq. 5 to solve linear systems on the UCI *protein* dataset in half precision, varying the size of the preconditioner directly. We find that the computations in Eq. 5 accumulate a large number of run-off errors which results in high residual norms of the resulting solves as seen in Figure 3.

## 5 OUR METHOD

Our methodology expands upon the ideas of Gardner et al. [2018] and Wang et al. [2019] for training GPs solely through matrix-vector operations via CG. We modify the CG algorithm in order to support half precision by increasing the numerical stability through: scale translation, mixed precision aggregation and re-orthogonalizaton.

### 5.1 HALF PRECISION MATRIX VECTOR MULTIPLIES

Following Wang et al. [2019] and Meanti et al. [2020], we exploit KeOps [Charlier et al., 2021] to produce our *matrix-free* approach. Where *matrix-free* refers to the numerical strategy of evaluating matrix multiplies by generating, on the fly, the entries required by the operation without the need to hold the matrix in memory. For a more detailed explanation of KeOps, please see Charlier et al. [2021] and Feydy et al. [2020]. More specifically, to use conjugate gradients effectively, we need to evaluate $\tilde{\mathbf{K}}\mathbf{v}$ which is written as

$$\tilde{\mathbf{K}}\mathbf{v}_i = a^2 \sum_{j=1}^{N} k\left(x_i, x_j\right) v_j + \sigma^2 v_i. \tag{4}$$

KeOps performs the inner loop of this multiplication via partitioning the resulting rows and columns of the matrix into blocks $B_1, \cdots B_k$ before reducing the resulting matrices, computing the row-wise operations in parallel. To exploit parallelism on accelerated hardware, we decompose the data

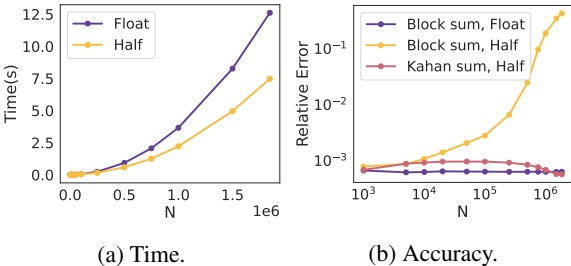

(a) Time.  (b) Accuracy.

Figure 4: **(a)** Timings for a single precision matrix vector multiplication on the HouseElectric dataset. Half is consistently about twice as fast as single precision. **(b)** Accuracies compared to the single precision for various schemes. Block summation performs each block within half precision, before doing accumulations in half or single and then casting the result back to half, while Kahan summation performs all operations in half. Block summation with single and Kahan summation are nearly as accurate as an entirely single precision MVM.

into blocks $B_1, \ldots, B_K$ of size $|B_k| = M << N$ ($M$ is the CUDA block size of 192) such that $\bigcup_{k=1}^K B_k = \{1, \ldots, N\}$. Moreover, we decompose $\mathbf{Kv}$ into $K$ separate products $\mathbf{K}^{(k)}\mathbf{v}^{(k)}$ where the $\mathbf{K}^{(k)}$ is an $M \times M$ matrix composed of $\mathbf{K}_{i,j}^{(k)} = a^2 k(x_i, x_j) + \sigma^2 \delta_{i,j}$ for all $i, j \in B_k$ and where $\mathbf{v}_i^{(k)} = \mathbf{v}_i$ for $i \in B_k$. We compute each separate product $\mathbf{K}^{(k)}\mathbf{v}^{(k)}$ as a regular matrix vector product (MVM). We note that a *matrix-free* approach does involve the creation of block matrices but not of the full matrix $\mathbf{K}$ where this approach only ever requires the storage of $M \times M$ block matrices at once. For accurate matrix products, we use block summation in a higher precision as is common on GPUs [Zamirai et al., 2021, NVIDIA, 2017].

To further exploit parallelism and to reduce memory overhead, we compute all matrix vector products in half precision, rather than single point precision [Gardner et al., 2018] or double precision. Moving to half precision immediately produces about a 2x speedup compared to half precision, as shown in Figure 4a, as evaluated on dataset sizes successively larger on the 1.8 million data point HouseElectric dataset from the UCI repository [Dua and Graff, 2017]. Furthermore, as shown in Figure 4b, these matrix vector multiplications are highly accurate with relative errors averaging less than 0.1% when we perform block summation in single precision. By comparison, block summation in half precision is significantly less accurate, while Kahan summation [Kahan, 1965] in half is nearly as accurate as block summation in single precision. All three approaches have similar speeds as shown in the Appendix. Thus, by moving to half precision we have immediately achieved a nearly 2x speedup in time complexity (and a similar reduction in memory) at negligible accuracy costs.

However, the limitations of the range can cause numerical

**Algorithm 1** Enhanced Stability CG (blue font denotes differences from standard CG)

1: **Input:** MVM function $\mathbf{K}(\cdot)$, initial solution guess $\mathbf{x}_0$, linear system right hand side $\mathbf{b}$, tolerance $\epsilon$, preconditioner function $\mathbf{P}(\cdot)$
2: **Initialize:** $k \leftarrow 0$, $\mathbf{r}_0 \leftarrow \mathbf{K}(x_0) - \mathbf{b}$, $\mathbf{d}_0 \leftarrow -\mathbf{r}_0$, $\mathbf{z}_0 = \mathbf{P}(\mathbf{r}_0)$ and $\log \gamma_0 \leftarrow \mathrm{L\Sigma E}(\mathbf{r}_0^T \mathbf{z}_0)$.
3: **while** $\|\mathbf{r}_k\|_2 < \epsilon$ **do**
4:     $\alpha_k = \exp\left(\log \gamma_k - \mathrm{L\Sigma E}\left(\mathbf{d}_k^T \mathbf{K}(\mathbf{d}_k)\right)\right)$
5:     $\mathbf{x}_{k+1} = \mathbf{x}_k + \alpha_k \mathbf{d}_k$
6:     $\mathbf{r}_{k+1} = \mathbf{r}_k + \alpha_k \mathbf{K}(\mathbf{d}_k)$
7:     **for** $j = 0$ to $k$ **do**
8:         $\mathbf{r}_{k+1} = \mathbf{r}_{k+1} - \left(\mathbf{u}_j^T \mathbf{r}_{k+1}\right)\mathbf{u}_j$
9:     **end for**
10:     $\mathbf{z}_{k+1} = \mathbf{P}(\mathbf{r}_{k+1})$
11:     $\log \gamma_{k+1} = \mathrm{L\Sigma E}\left(\mathbf{r}_{k+1}^T \mathbf{z}_{k+1}\right)$
12:     $\beta_{k+1} = \exp\left(\log \gamma_{k+1} - \log \gamma_k\right)$
13:     $\mathbf{d}_{k+1} = -\mathbf{r}_{k+1} + \beta_{k+1}\mathbf{d}_k$
14: **end while**

overflow for large matrix vector multiplications. To see why this is problematic, we note that the largest value representable in half precision is $2^{13}$, while the output of a kernel matrix vector multiply scales quasi-linearly with the size of the kernel matrix, $N$,

$$
\tilde{\mathbf{K}}(\mathbf{v})_i = a^2 \sum_{j=1}^N k(x_i, x_j) v_j + \sigma^2 v_i.
$$
$$
\leq N\left(a^2 + \sigma^2\right)\|v\|_\infty.
$$

We instead downscale every MVM by $N^{-1/2}$, computing $\tilde{\mathbf{K}}(\mathbf{v}/N^{1/2})$, which produces an upper bound that scales as $N^{1/2}$.

### 5.2 HALF PRECISION CONJUGATE GRADIENTS

**Rescaling Conjugate Gradients** The step sizes of the CG algorithm have a natural interpretation: $\alpha_k$ ensures we are minimizing the objective function along the path $\mathbf{x}_k + \alpha_k \mathbf{p}_k$ and $\beta_k$ guarantees conjugacy between the search directions. The accelerated convergence of CG depends on the correct computation of $\alpha_k$ and $\beta_k$; however, in finite arithmetic we cannot compute these quantities exactly. Worse, the round-off error is amplified when using half precision as we compute the step size $\alpha_k$ and $\beta_k$ terms

$$
\alpha_k = \frac{\mathbf{z}_k^T \mathbf{r}_k}{\mathbf{d}_k^T \mathbf{K}\mathbf{d}_k} \quad \text{and} \quad \beta_{k+1} = \frac{\mathbf{z}_{k+1}^T \mathbf{r}_{k+1}}{\mathbf{z}_k^T \mathbf{r}_k}.
$$

We need to prevent round-off and overflow error with these terms, so we store them solely in the log-scale (as they are positive by definition). To do so, we exploit the well known logsumexp trick, which states that for a given vector $\mathbf{w}$

and $\mathbf{z}$ such that $\mathbf{w}^T \mathbf{z} > 0$, to compute $\log \mathbf{w}^T \mathbf{z}$ we use the following transformation $\mathrm{L\Sigma E}(\cdot)$ such that

$$\mathrm{L\Sigma E}\left(\mathbf{w}^T \mathbf{z}\right) = y_{\max} + \log\left(\sum_{i=1}^{N} s_i \exp\left(y_i - y_{\max}\right)\right)$$

where $y_i = \log|w_i| + \log|z_i|$ and $s_i = \text{sign}(w_i z_i)$. For example, we compute and store $\log \alpha_k = \mathrm{L\Sigma E}(\mathbf{r}_k^\top \mathbf{z}_k) - \mathrm{L\Sigma E}(\mathbf{d}_k^\top \mathbf{K d}_k)$.

**Re-orthogonalization**    In infinite precision, each direction vector $\mathbf{d}_j$ for CG is $\mathbf{K}$ orthogonal, that is $\mathbf{d}_i^T K \mathbf{d}_j = 0$ whenever $i \neq j$, producing residual vectors, $\mathbf{r}_k$, that are $\mathbf{K}$ orthogonal to each other [Nocedal and Wright, 2006]. In practice, orthogonality is lost due to round-off error leading to slower convergence or even divergences in the residual vectors [Gratton et al., 2021, Cutajar et al., 2016]. To accelerate convergence and preserve orthogonality, we apply explicit Gram-Schmidt re-orthogonalization inside CG [Gratton et al., 2021]. Specifically, we re-orthogonalize the residual vector for time step $k$, updating the new residual $\mathbf{r}_k = \mathbf{K x}_k - \mathbf{b}$ as $\mathbf{r}_{k+1} \leftarrow \mathbf{r}_{k+1} - (\mathbf{u}_i^T \mathbf{r}_{k+1})\mathbf{u}_i$ for $i = 1, \ldots, k$ and where $\mathbf{u}_i$ are the previous iteration's orthonormal vectors created from the residual $\mathbf{r}_i$. Re-orthogonalization comes at an increased memory cost, as we must store all previous residuals, adding $\mathcal{O}(JN)$ memory costs for $J$ steps of CG. However, we find that with re-orthogonalization (and high tolerances) we are able to converge in very few steps, often $J < 50$.

**Preconditioning**    With preconditioning, we hope to find an operator such that $\mathbf{P}^{-1}\mathbf{K} \approx \mathbf{I}$ with $\mathbf{P}^{-1}(\mathbf{v})$ is inexpensive to compute and solve the system $\mathbf{P}^{-1}\mathbf{K v} = \mathbf{P}^{-1}\mathbf{w}$ instead of $\mathbf{K v} = \mathbf{w}$. Following Gardner et al. [2018], we use the pivoted Cholesky decomposition, which requires solely access to the rows of the matrix (e.g. $\mathbf{K}_{i\cdot} = \mathbf{K}e_i$ where $e_i$ is a zero vector with $i$th entry equal to one) and an approximate diagonal (constant in our case). Running $k$ steps of the pivoted Cholesky decomposition on $\mathbf{K}$ produces an approximation $\tilde{\mathbf{K}} \approx \mathbf{L}_k \mathbf{L}_k^\top + \sigma^2 I$ and we use the Woodbury matrix identity to construct $\mathbf{P}^{-1}$.

$$\mathbf{P}^{-1}\mathbf{w} = \sigma^{-2}\mathbf{w} -$$
$$\sigma^{-4}\mathbf{L}_k \left(\mathbf{I} + \sigma^{-2}\mathbf{L}_k^T \mathbf{L}_k\right)^{-1} \mathbf{L}_k^T \mathbf{w}. \quad (5)$$

The inner system is of size $k << N$ and so can be solved using direct methods (e.g. Cholesky or QR) in single precision (due to a lack of LAPACK support for these in half).

To summarize, we show our revised CG procedure in algorithm 1, with the differences from the BBMM CG algorithm of Gardner et al. [2018] in bolded blue font.

### 5.3  LOSS FUNCTION

Computation of Eq. (1) requires either an expensive eigendecomposition or Lanczos iteration in the forwards pass to

compute the log determinant term, which can be quite unstable [Gardner et al., 2018]. Instead, we use a pseudo-loss that has the same gradient as Eq. (2) via solving $\tilde{K}$ against both the response $\mathbf{y}$ and the probe vectors $\mathbf{z}_i$ simultaneously. That is, we find solutions to the linear system using Alg. 1:

$$\mathbf{K}\left[\mathbf{u}_0, \mathbf{u}_1, \cdots, \mathbf{u}_M\right] = \left[\mathbf{y}, \mathbf{z}_1, \cdots, \mathbf{z}_M\right]. \quad (6)$$

for $\mathbf{u}_i$. We then detach these solutions and compute simply:

$$\tilde{\mathcal{L}}(\boldsymbol{\theta}) = \frac{1}{2M}\sum_{j=1}^{M} \mathbf{u}_j^T\left(\mathbf{K}_{\boldsymbol{\theta}}\mathbf{z}_j\right) - \frac{1}{2}\mathbf{u}_0^T\left(\mathbf{K}_{\boldsymbol{\theta}}\mathbf{u}_0\right), \quad (7)$$

which has gradient

$$\nabla_{\boldsymbol{\theta}}\tilde{\mathcal{L}} = \frac{1}{2M}\sum_{j=1}^{M} \mathbf{u}_j^T\left(\nabla_{\boldsymbol{\theta}}\mathbf{K}\mathbf{z}_j\right) - \frac{1}{2}\mathbf{u}_0^T\left(\nabla_{\boldsymbol{\theta}}\mathbf{K}\mathbf{u}_0\right), \quad (8)$$

which is the same as Eq. 2. Eq. 8 then only requires gradients with respect to matrix vector products, which are computed natively in KeOps. Eq. (8) does not use variance reduction strategies such as Wenger et al. [2021] but they could also be incorporated.

The computational complexity of these operations is the same as that of computing the standard log marginal likelihood [Gardner et al., 2018]. Computing Eq. 8 requires $\mathcal{O}((M+1)N^2)$ time where $M$ is the number of trace estimates and $N$ is the total number of data points. Using half precision does not alter the order of the cost but only reduces the constant of the MVMs by half.

## 6  BENCHMARKING EXPERIMENTS

We compare our method directly to single precision GP training using KeOps and GPyTorch on benchmarks of up to 1.8 million data points from the UCI repository [Dua and Graff, 2017] using RBF ARD kernels and a single NVIDIA V100 GPU. We show results for other kernels in Appendix C. A mixed precision strategy of using solely half precision matrix vector multiplications produces speedups of at least two times over an equivalent single precision model.

**Residual Norms**    First, we consider the relative solve error on three different datasets of varying size: *Elevators*, *KeggDirected*, and *Buzz* for hyperparameters collected at the end of the optimization runs. While standard CG in half precision diverges very quickly, we find that our stable CG implementation in half precision closely matches the convergence behaviour of single precision CG, converging to a residual tolerance less than $0.5$ in less than $50$ optimization steps, as shown in Figures 5a for *elevators* and 5b for *KeggDirected*.

Furthermore, we explore the effects of using no preconditioner (rank 0) and also of using preconditioners of rank 5,

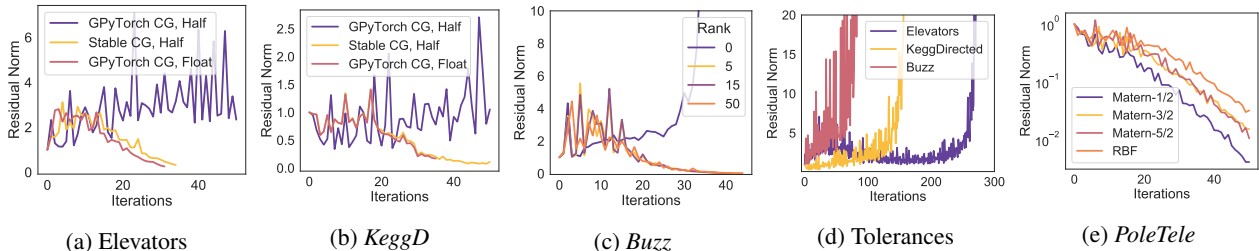

| | (a) Elevators | (b) *KeggD* | (c) *Buzz* | (d) Tolerances | (e) *PoleTele* |

Figure 5: **(a,b)** Residual norms on *Elevators* and *KeggD*; standard half precision CG fails while our approach converges in a similar number of iterations to single precision CG. **(c)** Residual norms using our CG solver on *Buzz*; using preconditioning prevents divergences due to round-off errors. **(d)** Running CG for too many iterations produces round-off errors that build up over time producing divergences. **(e)** Matérn-$1/2, 3/2$ kernels tend to converge quicker than RBF kernels on *PoleTele*.

Table 1: RMSEs and training time with $\pm$ the standard deviation of the results over 5 different seeds on a suite of UCI tasks for half and single precision GPs and SVGPs. Here, we use RBF ARD kernels with 50 CG iterations and 50 optimization steps. We included KeOps compilation times (which can be significant, see Table 1) and observe that the largest improvements come on *Song* and *HouseElectric*.

| | | RMSE | | | Time | | |
|---|---|---|---|---|---|---|---|
| Dataset | $(N, d)$ | Single | Half | SVGP | Single | Half | SVGP |
| PoleTele | (13.5K, 26) | $0.117 \pm 0.004$ | $0.121 \pm 0.003$ | $0.14 \pm 0.004$ | $57.6 \pm 0.6$ | $88.3 \pm 3.2$ | $15. \pm 0.5$ |
| Elevators | (14.9K, 18) | $0.364 \pm 0.004$ | $0.382 \pm 0.001$ | $0.374 \pm 0.005$ | $58.4 \pm 3.1$ | $90.1 \pm 2.9$ | $16 \pm 0.5$ |
| Bike | (15.6K, 17) | $0.074 \pm 0.003$ | $0.083 \pm 0.009$ | $0.077 \pm 0.006$ | $62.3 \pm 0.7$ | $85.6 \pm 0.41$ | $17 \pm 0.6$ |
| Kin40K | (36K, 8) | $0.099 \pm 0.001$ | $0.100 \pm 0.003$ | $0.165 \pm 0.003$ | $65.5 \pm 5.3$ | $92.9 \pm 0.28$ | $39 \pm 0.5$ |
| Protein | (41.1K, 9) | $0.055 \pm 0.006$ | $0.635 \pm 0.002$ | $0.632 \pm 0.01$ | $70.9 \pm 5.7$ | $115.7 \pm 0.10$ | $46 \pm 3.2$ |
| 3droad | (391.4K, 3) | $0.194 \pm 0.01$ | $0.215 \pm 0.004$ | $0.412 \pm 0.011$ | $1,260 \pm 35$ | $1,003 \pm 0.97$ | $412 \pm 101$ |
| Song | (463.8K, 90) | $0.761 \pm 0.004$ | $0.779 \pm 0.022$ | $0.999 \pm 0.002$ | $24,357 \pm 2,613$ | $9,930 \pm 130$ | $284 \pm 2.0$ |
| Buzz | (524.9K, 77) | $0.300 \pm 0.01$ | $0.448 \pm 0.225$ | $0.268 \pm 0.004$ | $25,436 \pm 1,200$ | $23,127 \pm 1,819$ | $617 \pm 114$ |
| HouseElectric | (1844.3K, 9) | $0.052 \pm 0.002$ | $0.051 \pm 0.004$ | - | $42,751 \pm 2,180$ | $36,025 \pm 1,178$ | - |

15 and 50 on *Buzz*. As seen in Figure 5c, we find that CG diverges without preconditioning and that a rank 5 preconditioner is sufficient for convergence and behaves similarly to preconditioners with rank 15 and 50. These findings are consistent with Maddox et al. [2021].

Despite preconditioning, the effect of low-precision does produce round-off errors that grow with the number of iterations, preventing us from running our method for a large number iterations or with low tolerances. We show this effect across three datasets in Figure 5d. Finally, we show the results across Matérn kernels with varying $\nu$ and RBF kernels without ARD on *PoleTele* in Figure 5e, finding that Matérn-$1/2$ kernels tend to converge fastest. This is consistent with our discussion in Section 4.2, since the Matérn-$1/2$ kernels have a wider range of numerically representable distances, reducing round-off error and they also have better conditioning.

**Optimization Trajectories** Next, in Figure 6, we display the evolution of the three hyperparameters (noise, outputscale, and lengthscale) on *Protein* with the same plot shown for *3droad* in Figure A.5. We see that the solver produces accurate enough solves so that gradients are unaffected and the hyperparameters train similarly to the GPyTorch solves in single precision.

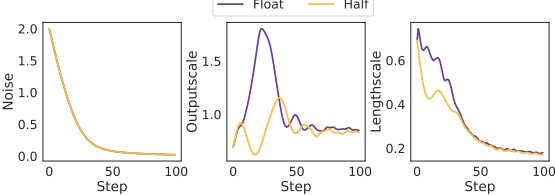

Figure 6: Optimization and parameter trajectories on *Protein* for both a standard single precision model and our half precision implementation. The noise term (left) and lengthscale (right) are very close for both precisions, while the outputscale (center) shows some minor differences, but convergence to a similar value.

**Benchmark Results** Finally, in Table 1 we display the root mean square errors (RMSEs) and fitting times for both single and half precision as well as variational GPs (SVGPs) [Hensman et al., 2013] over 5 seeds, displaying the mean and standard deviation. We see that it takes up 100k datapoints for there to be a significant speedup from using half precision, which is due to to increased compilation times in KeOps in half precision (see Table 1 in the Appendix). However, for *Song* and *HouseElectric*, the two largest datasets we see significant speedups (of up to 3x) for the same number of optimization steps. Furthermore, SVGPs tend to be faster but often perform significantly worse than half or

single precision GPs. We were unable to run SVGPs on *HouseElectric* due to out of memory errors at test time on our servers.

# 7 CONCLUSION

Deep learning has benefited enormously from advances in hardware design and systems. While GPs have started to exploit GPU parallelization, with highly impactful systems such as GPyTorch [Gardner et al., 2018], there is essentially no work on low-precision inference with GPs, despite the widespread impact of low-precision optimization in deep learning. Indeed, low-precision GPs are nontrivial because the operations necessary for inference are relatively sensitive to the precision of the computations. In this respect, our work provides several new insights:

- *Training GPs in low-precision requires algorithmic modifications to ensure numerical stability*. To this end, we propose a stable version of conjugate gradients that uses scale translation, re-orthogonalization, preconditioning and mixed precision.

- *Mixing the precision of computations guarantees speedups while minimizing issues related to round-off errors*. To minimize catastrophic round-off error it is important to cast inexpensive but critical computations, such as the step sizes in CG, into higher precisions, while performing noncritical and large computations, such as MVMs, on lower precisions to gain runtime speedups.

- *The kernel choice affects the convergence behavior of GPs in lower precisions*. Mátern kernels converge faster than RBF kernels, as Mátern kernels have a wider range of numerically representable distances, reducing round-off error and poor conditioning.

- *Lower precisions slow the rate of decay of the eigenspectrum of the kernel matrices*. The change in the rate of decay of the eigenvalues introduced by using lower precisions has a detrimental effect both on the training convergence, by slowing the rate of progress of the CG iterations, and on the provable generalization guarantees, by increasing the effective dimensionality.

- *How the bits are split between matissa and exponent affect the training results*. We find float16 to be the easiest half precision standard to work with since it has a wider range of values (when compared to bfloat16) which avoids the CG steps from being clipped.

- *Our algorithm for training GPs in lower precisions can be used to broadly accelerate standard CG, beyond Gaussian process inference*. In many cases, there is no reason not to use the proposed procedures as a drop-in replacement for standard CG. Apart from re-orthogonalization, the rest of the modifications that we

introduced into CG have no effect on single or double precision, and apply broadly.

These contributions will continue to appreciate with time, as hardware advances continue to further accelerate low-precision computations.

There are also many promising directions for future research. The use of lower precisions and our mixed precision approach can be used in other iterative methods such as Lanczos [Pleiss et al., 2018] and MINRES [Pleiss et al., 2020]. Our methods could also be adapted to approximate GP inference, e.g., for classification, and combined with scalable low rank approximations of the kernel matrix.

**Acknowledgements** We would like to thank Chris De Sa and Ke Alexander Wang for helpful discussions. This research is supported by an Amazon Research Award, Facebook Research, Google Research, Capital One, NSF CAREER IIS-2145492, NSF I-DISRE 193471, NIH R01DA048764-01A1, NSF IIS-1910266, and NSF 1922658 NRT-HDR.

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
