# OpenReview forum: "Low-Precision Arithmetic for Fast Gaussian Processes"
_auai.org/UAI/2022/Conference — UAI 2022 Poster_

### Official Review · Reviewer_MSfS · 2022-04-10

**Q2(1) Originality/Novelty:** 3
**Q2(2) Significance/Impact:** 3
**Q2(3) Correctness/Technical Quality:** 3
**Q2(6) Clarity Of Writing:** 3
**Q6 Overall Score:** 4
**Q8 Confidence In Your Score:** 4

**Q1 Summary And Contributions:**

The paper uses low precision floating points in the conjugate gradient approach to learning Gaussian processes. A number of numerical techniques are combined in section 4.2 to give a working implementation that is illustrated in the experiments.

**Q2 Assessment Of The Paper:**

More detailed information regarding each of these aspects is given below:

**Q2(4) Quality Of Experiments (Optional):**

3: Good: The experimental evaluation is adequate, and the results convincingly support the main claims.

**Q2(5) Reproducibility:**

3: Good: Key resources (e.g., proofs, code, data) are available and key details (e.g., proofs, experimental setup) are sufficiently well-described for competent researchers to confidently reproduce the main results.

**Q3 Main Strengths:**

The paper managed to combine a few tricks in section 4.2 to make low-precision conjugate gradient (CG) work for learning Gaussian process. I think this is non-trivial, and is only obvious on the hind-sight.

The paper is easy to follow, and experimental results show convincing the speedups given by the method (up to 3x).

**Q4 Main Weakness:**

The main weakness is in the stated first contribution in section 1, which I find lacking in the rest of the paper. Section 5.1 should be another section all by itself and also includes considerations with respect to the theoretical aspects of GP and decay-rate of the eigen-values. For example, saying "RBF kernels have the smallest support, ..., followed by Matern kernels" totally mis-regard that the RBF gives process that is infinitely mean-square differentiable (i.e., actually has the largest supported theoretically, albeit more very weak influence from far-away points), while Matern gives a process that is finitely mean-square differentiable. Analysis of the eigen-spectral must include the theoretically implications.

**Q5 Detailed Comments To The Authors:**

- Initial-capitalized "Cholesky" instead of "cholesky" in multiple places.

- Pg 3, left column, third line from bottom: "wile" -> "while"

- Eq. 3: brackets around the MVM before the subscript.

- Pg 4, left column, last para is probably is miswrite.

- Algo 1: bolded blue font not useful in gray-scale printout.

- Section 5.1, first para: "converging" should be "decaying"?

- Section 5.1, last para: Very first work is in [15].

- Section 5.1.1: "mock" should be "replicate"?

- Section 5.2, "residual norms": "hyperparmeters" is a mispelling.

- Section 5.2, last para: "compilation times in KeOps in half" needs a rewrite.

[15] Manfred Opper and Francesco Vivarelli (1999). General bounds on Bayes errors for regression with
Gaussian processes. NIPS 11.


**Q7 Justification For Your Score:**

While this is an important practical work, the claimed contribution on eigen-spectrum analysis does not address the theoretical aspects of Gaussian processes.

**Q9 Complying With Reviewing Instructions:**

1: Yes.

---

### Official Review · Reviewer_p5gt · 2022-04-12

**Q2(1) Originality/Novelty:** 2
**Q2(2) Significance/Impact:** 3
**Q2(3) Correctness/Technical Quality:** 3
**Q2(6) Clarity Of Writing:** 3
**Q6 Overall Score:** 6
**Q8 Confidence In Your Score:** 3

**Q1 Summary And Contributions:**

The paper proposes a study on low precision arithmetic for reducing the computational costs of Gaussian process models. In particular the authors propose a variant of conjugate gradients which delivers a more stable convergence. The method is tested against full precision GPs and shows important time savings.

**Q2 Assessment Of The Paper:**

More detailed information regarding each of these aspects is given below:

**Q2(4) Quality Of Experiments (Optional):**

2: Fair: The experimental evaluation is weak: important baselines are missing, or the results do not adequately support the main claims.

**Q2(5) Reproducibility:**

3: Good: Key resources (e.g., proofs, code, data) are available and key details (e.g., proofs, experimental setup) are sufficiently well-described for competent researchers to confidently reproduce the main results.

**Q3 Main Strengths:**

- It is interesting to see how, with an appropriate treatment of the error, it is possible to use half precision matrices without losing too much in performance
- The authors do a good job explaining the steps involved with half precision conjugate gradients

**Q4 Main Weakness:**

- A comparison with one of the sparse inducing points methods could put in perspective how good is the performance of low precision GP.

**Q5 Detailed Comments To The Authors:**

- As mentioned above, how does the half precision GP perform compared to other approximation methods, such as sparse inducing points methods?
- Could the authors add more details on the computation of the loss function? In particular, how expensive is eq. 5?
- More in general, how is this method used to compute the posterior covariance at a test point? Do we need an inversion for each test point?

**Q7 Justification For Your Score:**

The use of low precision in GPs is very interesting and could be potentially powerful.

**Q9 Complying With Reviewing Instructions:**

1: Yes.

---

### Official Review · Reviewer_zR2g · 2022-04-13

**Q2(1) Originality/Novelty:** 3
**Q2(2) Significance/Impact:** 3
**Q2(3) Correctness/Technical Quality:** 3
**Q2(6) Clarity Of Writing:** 4
**Q6 Overall Score:** 6
**Q8 Confidence In Your Score:** 4

**Q1 Summary And Contributions:**

The authors describe a set of techniques to have GPs work in half precision on GPUs, which results in a speedup in computations. To circumvent the problems of working with half precision they proposed a method based on conjugate gradients with re-orthogonalization, compact kernels and preconditioners. The method improves numerical stability and allows to run GPs on 1.8M data points without sparse approximations.

**Q2 Assessment Of The Paper:**

More detailed information regarding each of these aspects is given below:

**Q2(4) Quality Of Experiments (Optional):**

3: Good: The experimental evaluation is adequate, and the results convincingly support the main claims.

**Q2(5) Reproducibility:**

3: Good: Key resources (e.g., proofs, code, data) are available and key details (e.g., proofs, experimental setup) are sufficiently well-described for competent researchers to confidently reproduce the main results.

**Q3 Main Strengths:**

- Significant engineering effort to make GPs work in half precision.
- A new conjugate gradient based method that has improved stability.
- Detailed experiments illustrating the usefulness of the proposed methods.
- Well written paper.
- Relevant problem: how to make GPs work well with low precision.



**Q4 Main Weakness:**

- The computational gains seem to be not so high in some cases, e.g. in House Electric dataset with 1.8M data points the cost is only 80% of the single precision.


**Q5 Detailed Comments To The Authors:**

Why is it that Float did not converge in Buzz? It seems strange.

**Q7 Justification For Your Score:**

This is a well-written paper that addresses a relevant problem. The contributions are a bunch of engineering tricks to make things
work, but as a whole, they are very valuable.


**Q9 Complying With Reviewing Instructions:**

1: Yes.

---

### Official Review · Reviewer_284r · 2022-04-19

**Q2(1) Originality/Novelty:** 3
**Q2(2) Significance/Impact:** 3
**Q2(3) Correctness/Technical Quality:** 3
**Q2(6) Clarity Of Writing:** 4
**Q6 Overall Score:** 7
**Q8 Confidence In Your Score:** 3

**Q1 Summary And Contributions:**

Gaussian Processes are powerful but computationally expensive to fit. The alternative is to use sparse GPs, but this paper presents an interesting alternative: a reformulation which leverages mixed-precision arithmetic, empirically demonstrated to significantly save computation time.

**Q2 Assessment Of The Paper:**

More detailed information regarding each of these aspects is given below:

**Q2(4) Quality Of Experiments (Optional):**

4: Excellent: The experimental evaluation is comprehensive and the results are compelling.

**Q2(5) Reproducibility:**

3: Good: Key resources (e.g., proofs, code, data) are available and key details (e.g., proofs, experimental setup) are sufficiently well-described for competent researchers to confidently reproduce the main results.

**Q3 Main Strengths:**

- the implementation is well-documented (PyTorch code)
- the clear and modular theoretical analysis is provided
- fits the recent efforts of the communities around AD frameworks on leveraging mixed-precision arithmetic

**Q4 Main Weakness:**

- from the theory perspective, this is not a major breakthrough on the time/space complexity, as making this work we gain only (as much as?) a solid constant. However I acknowledge this improves scalability in practice and can be boosted further on certain hardware.

**Q5 Detailed Comments To The Authors:**

- please clarify which optimisation aspects would be new to PyTorch and which could be recycled already. I know more Tensorflow and there there are already some tricks on mixed-precision implemented, so there (TF) not everything needs to be written from scratch.
- what are plans to go beyond the framework of Gaussian Processes?

**Q7 Justification For Your Score:**

This is a nicely documented optimisation attempt to the task of fitting GPs. While it doesn't solve the "big" complexity problem in asymptotical terms, it has a practical value and nicely fits the recent research.

**Q9 Complying With Reviewing Instructions:**

1: Yes.

---

### Decision · Program_Chairs · 2022-05-15

**Decision:**

Accept (Poster)

**Comment:**

Meta Review: This submission focuses on numerical precision issues for Gaussian Processes (GPs). In particular, the focus is on training GPs in half-precision. The paper is not theoretical. The authors shows (numerically) that half precision kernel matrices have qualitatively different eigenspectra than higher precision representations, and represent covariances at much shorter distances than in float or double precision. This is an interesting contribution.

Rebuttal: The authors have replied to reviewers' comments.

Suggested minor correction for the authors: clarify in the Introduction that some of the contributions were verified numerically, example change "We show that half precision kernel matrices ..." to "We numerically show that half precision kernel matrices ..."